# Interleukin-34-CSF1R Signaling Axis Promotes Epithelial Cell Transformation and Breast Tumorigenesis

**DOI:** 10.3390/ijms22052711

**Published:** 2021-03-08

**Authors:** Muna Poudel, Garam Kim, Poshan Yugal Bhattarai, Jin-Young Kim, Hong Seok Choi

**Affiliations:** College of Pharmacy, Chosun University, Gwangju 61452, Korea; munapaudel35@gmail.com (M.P.); garam@chosun.ac.kr (G.K.); poshanb@chosun.kr (P.Y.B.); kkk423@hanmail.net (J.-Y.K.)

**Keywords:** interleukin-34, colony stimulating factor 1 receptor, activator protein-1, peptidyl-prolyl cis-trans isomerase NIMA-interacting 1

## Abstract

IL-34 has been recently identified as a ligand for CSF1R that regulates various cellular processes including cell proliferation, survival, and differentiation. Although the binding of IL-34 to CSF1R modulates several cancer-driving signaling pathways, little is known about the role of IL-34/CSF1R signaling in breast cancer. Herein, we report that IL-34 induces epithelial cell transformation and breast tumorigenesis through activation of MEK/ERK and JNK/c-Jun pathways. IL-34 increased the phosphorylation of MEK1/2, ERK1/2, JNK1/2, and c-Jun through CSF1R in mouse skin epidermal JB6 C141 cells and human breast cancer MCF7 cells. IL-34 enhanced c-Fos and c-Jun promoter activity, resulting in increased AP-1 transactivation activity in JB6 Cl41 and MCF7 cells. Moreover, PIN1 enhanced IL-34-induced phosphorylation of MEK1/2, ERK1/2, JNK1/2, and c-Jun in JB6 Cl41 and MCF7 cells. Inhibition of PIN1 using juglone prevented the IL-34-induced transformation of JB6 C141 cells. Similarly, silencing of PIN1 reduced the IL-34-induced tumorigenicity of MCF7 cells. Consistent with these results, the synergistic model showed that treatment with juglone suppressed the IL-34-induced growth of tumors formed by 4T1 cells in BALB/c mice. Our study demonstrates the role of IL-34-induced MEK/ERK and JNK/c-Jun cascades in breast cancer and highlights the regulatory role of PIN1 in IL-34-induced breast tumorigenesis.

## 1. Introduction

The tumor microenvironment (TME) has profound influence on tumor cell proliferation, metastasis, and angiogenesis [1]. TME is the area surrounding the tumor cells that is composed of immune cells, endothelial cells, fibroblasts, in addition to extracellular matrix proteins and signaling molecules produced by these cells [2,3]. Immune cells present in the microenvironment, such as macrophages, neutrophils, tumor-infiltrating lymphocytes, and T cells produce a wide range of signaling molecules collectively known as cytokines that play a role in the development and progression of cancers [4]. Interleukins (ILs) are a group of cytokines with complex immunomodulatory functions, including cell proliferation, migration, and adhesion [5]. ILs are known to have both pro-tumorigenic and anti-tumorigenic effects. For instance, IL-2, IL-15, and IL-21 are reported to promote tumor suppression in melanoma, whereas IL-1, IL-4, and IL-6 are associated with tumor progression in breast, prostate, and lung cancer [6]. Recent studies have shown that IL-34 expression is upregulated in several cancers [7]. In osteosarcoma, IL-34 has been shown to promote tumor progression and metastasis by recruiting tumor-associated macrophages (TAMs) [8]. IL-34 also promotes colorectal cancer cell growth and invasion [9]. It has also been demonstrated that IL-34 regulates the migration of breast cancer cells [10]. However, the role of IL-34 in breast carcinogenesis is poorly understood.

IL-34 is produced by different cell populations such as macrophages, adipocytes, endothelial cells, epithelial cells, and fibroblasts [7]. IL-34 synthesis is enhanced by pro-inflammatory cytokines, such as tumor necrosis factor-α (TNF-α) and IL-1β [11]. IL-34 is highly associated with cell proliferation, differentiation, apoptosis, angiogenesis, inflammation, and immunoregulation [8,12,13,14]. There is an increase in the serum levels of IL-34 under pathological conditions such as rheumatoid arthritis, systemic lupus erythematosus, Sjögren’s syndrome, systemic sclerosis, and cancer [15,16,17,18]. IL-34 promotes tumor growth, metastasis, immune suppression, and therapeutic resistance [8,19,20]. It mediates its function through interaction with colony stimulating factor 1 receptor (CSF1R) [21]. CSF1R is a receptor tyrosine kinase that is activated upon binding to its ligands IL-34 and CSF-1 through oligomerization and auto-phosphorylation [22]. Binding of IL-34 to CSF1R leads to the activation of various signaling pathways such as the nuclear factor κ-light-chain-enhancer of activated B cells (NF-κB) pathway, phosphoinositide 3-kinase (PI3K)/Akt, mitogen-activated protein kinase (MAPK), and Janus kinase (JAK)/signal transducer and activator of transcription (STAT)3 pathways, which are important for tumor development [21,22,23,24,25]. However, the specific signaling pathway regulated by IL-34 during breast tumor development and its functions in those pathways remain to be elucidated.

PIN1 is a peptidyl prolyl cis/trans isomerase (PPIase), which regulates the conformational changes around proline amino acid followed by phosphorylated serine or threonine [26]. PIN1 consists of two domains: an N-terminal WW domain that interacts with specific phosphorylated-serine/threonine-proline (pSer/Thr-Pro) motifs and a C-terminal PPIase domain that isomerizes the pSer/Thr-Pro motifs [26]. Such conformational changes can have profound effects on the function of many PIN1 substrates, such as p53 [27], cyclin D1 [28], mitogen-activated protein kinase kinase 1 (MEK1) [29], and c-Jun [30] thereby playing important roles in many cellular events, such as cell cycle progression, differentiation, and tumorigenesis. PIN1 is overexpressed in breast cancer and cooperates with RAS signaling to increase the transcriptional activity of c-Jun for tumorigenesis [30]. Inhibition of PIN1 prevents breast cancer development induced by oncogenes such as Neu or Ras in mice [31]. In addition, PIN1 enhances the IL-22-induced ERK1/2 and JNK1/2 signaling pathways, leading to breast tumorigenesis [32]. Although many details of PIN1 function have been elucidated, it is largely unknown whether PIN1 is involved in the IL-34 signaling pathway during breast tumorigenesis.

In the present study, we demonstrated that IL-34-induced MEK/ERK and JNK/c-Jun signaling is regulated by PIN1 in breast cancer. IL-34 increases the promoter activity of c-Fos and c-Jun, which eventually results in transactivation of AP-1. Furthermore, we showed that IL-34-induced neoplastic cell transformation is enhanced by PIN1. Our study not only illustrates the oncogenic function of IL-34 but also highlights the regulatory role of PIN1 in IL-34-induced breast tumorigenesis.

## 2. Results

### 2.1. IL-34 Induces Epithelial Cell Transformation and Mammary Gland Tumorigenesis

The JB6 Cl41 cell system is a well-developed model for studying tumor promotion under anchorage-independent growth conditions [33]. To determine whether IL-34 affects the cell proliferation and transformation of JB6 Cl41 cells, we initially examined the effects of IL-34 on the proliferation of JB6 Cl41 cells using a BrdU incorporation assay. We found that IL-34 treatment significantly and dose-dependently induced the proliferation of JB6 Cl41 cells (Figure 1A).

The cells were then subjected to a soft agar assay in the presence or absence of IL-34. The results showed that dose-dependent treatment with IL-34 not only increased the colony numbers but also the colony sizes in JB6 Cl41 cells (Figure 1B). Next, we examined the effects of IL-34 on MCF7 breast cancer cell growth using a BrdU incorporation assay and a soft agar assay. IL-34 significantly increased the number of cells (Figure 1C) as well as the formation of colonies in MCF7 cells (Figure 1D). Similarly, there was a significant dose-dependent increase in the proliferation of 4T1 cells upon treatment with IL-34 (Figure 1E). Furthermore, the in vivo effects of IL-34 on tumor development were studied in a mouse model of 4T1 metastatic mouse breast carcinoma cells. 4T1 cells were separately injected into the mammary glands of BALB/c mice in the presence or absence of IL-34. Representative tumor images demonstrated that there was an increase in the breast tumor growth in mice treated with IL-34, as compared to those treated with phosphate buffered saline (Figure 1F). Collectively, these results suggest that IL-34 induces cell proliferation, anchorage-independent cell transformation, and breast mammary tumorigenesis in vitro and in vivo.

### 2.2. IL-34 Activates MEK/ERK and JNK/c-Jun Signaling Through CSF1R in JB6 Cl41 Cells

Previous studies have reported that CSF1R not only binds to its ligand macrophage colony-stimulating factor (MCSF) but also triggers the MAPK signaling pathway [34]. As IL-34 is also characterized as an MCSF twin cytokine [21], we next examined the effects of IL-34 on MEK/ERK and JNK/c-Jun signaling pathways. IL-34 clearly induced the phosphorylation of MEK1/2 and ERK1/2 (Figure 2A,B) along with JNK1/2 and c-Jun (Figure 2C,D) in a dose- and time-dependent manner. 

To further examine whether the IL-34-induced MEK/ERK and JNK/c-Jun pathways were mediated by CSF1R, we transfected JB6 Cl41 cells with mouse siRNA-control and siRNA-CSF1R and then treated them with IL-34. The results showed that there was a decrease in IL-34-induced phosphorylation of MEK1/2, ERK1/2, JNK1/2, and c-Jun in CSF1R-knockdown cells, as compared to control cells (Figure 2E). Moreover, treatment with PD98059, a specific inhibitor of MEK1/2 and SP600125, a JNK1/2 inhibitor, suppressed IL-34-induced phosphorylation of ERK1/2 (Figure 2F) and c-Jun (Figure 2G). Collectively, these results indicate that IL-34 stimulates the MEK/ERK and JNK/c-Jun signaling pathways via CSF1R in JB6 Cl41 cells.

### 2.3. PIN1 Regulates IL-34-Induced MEK/ERK and JNK/c-Jun Signaling in JB6 Cl41 Cells

Previous studies have reported that PIN1 interacts with MEK1 [35] and c-Jun [30]. In light of this, we examined whether PIN1 could affect the IL-34-induced MEK/ERK and JNK/c-Jun signaling pathways. JB6 Cl41 cells were transfected with mock and Xpress-PIN1, followed by treatment with or without IL-34. The results showed that IL-34-mediated phosphorylation of MEK1/2, ERK1/2, JNK1/2, and c-Jun were notably increased in PIN1-overexpressing cells, as compared to control cells (Figure 3A). To further confirm the regulatory role of PIN1 on IL-34-induced MEK/ERK and JNK/c-Jun signaling cascades, JB6 Cl41 cells were transfected with mouse siRNA-control and siRNA-PIN1, followed by treatment with or without IL-34. Knockdown of PIN1 suppressed IL-34-induced phosphorylation of MEK1/2, ERK1/2, JNK1/2, and c-Jun in JB6 Cl41 cells (Figure 3B). 

Next, JB6 Cl41 cells were treated with juglone, a potent PIN1 inhibitor, in the presence or absence of IL-34. Treatment with juglone inhibited IL-34-induced phosphorylation of MEK1/2, ERK1/2, JNK1/2, and c-Jun (Figure 3C). Altogether, these results illustrated that PIN1 regulates IL-34-induced MEK/ERK and JNK/c-Jun signaling in JB6 Cl41 cells.

### 2.4. PIN1 Regulates IL-34-Induced AP-1 Activity and Cell Transformation in JB6 Cl41 Cells

Given the increased phosphorylation of MEK1/2, ERK1/2, JNK1/2, and c-Jun by IL-34, we examined the effects of IL-34 on the transcriptional activity of AP-1 subunits c-Fos and c-Jun. The result showed that IL-34 treatment significantly increased c-Fos activity (Figure 4A). Similarly, c-Jun activity was also significantly upregulated upon treatment with IL-34 in JB6 Cl41 cells (Figure 4B). In addition, treatment with IL-34 caused a significant induction of AP-1 transactivation at higher dose (Figure 4C). Furthermore, to examine whether PIN1 is essential for IL-34-induced AP-1 activity and epithelial cell transformation, we first assessed IL-34-induced AP-1 activity in the presence or absence of juglone. The results showed that treatment with juglone remarkably inhibited the IL-34-induced increase in AP-1 activity (Figure 4D). Next, we performed soft agar colony assays and examined the effects of juglone on IL-34-induced anchorage-independent growth of JB6 Cl41 cells. The results showed that juglone significantly inhibited the transformation of JB6 Cl41 cells induced by IL-34. Collectively, our results suggest that PIN1 regulates AP-1 activity and IL-34-induced cellular transformation of JB6 Cl41 cells.

### 2.5. PIN1 Enhances IL-34-Induced MEK/ERK and JNK/c-Jun Signaling in Breast Cancer Cells

After determining the involvement of IL-34 in the activation of MEK/ERK and JNK/c-Jun signaling cascades in JB6 Cl41 cells, we further confirmed whether IL-34 influences the MAPK pathway in breast cancer cells. First, we examined the effects of IL-34 on the active phosphorylated forms of MEK1/2, ERK1/2, JNK1/2, and c-Jun in MCF7 and SKBR3 cells. We found that treatment with IL-34 increased the phosphorylation levels of MEK1/2, ERK1/2, JNK1/2, and c-Jun in MCF7 and SKBR3 cells in a dose- and time-dependent manner (Figure 5A,B). 

Next, to determine whether knockdown of CSF1R affects IL-34-induced phosphorylation of MEK1/2, ERK1/2, JNK1/2, and c-Jun, MCF7 cells were transfected with siRNA-control and siRNA-CSF1R, followed by treatment with IL-34. The results showed that IL-34-induced MEK1/2, ERK1/2, JNK1/2, and c-Jun phosphorylation was suppressed in CSF1R-knockdown cells, as compared to control cells (Figure 5C). Furthermore, to understand the role of PIN1 in the IL-34-induced MAPK pathway in breast cancer, we transfected the cells with mock and Xpress-PIN1 or siRNA-control and siRNA-PIN1, followed by treatment with or without IL-34 in MCF7 cells. The results showed that IL-34-induced phosphorylation of MEK1/2, ERK1/2, JNK1/2, and c-Jun was increased in PIN1-overexpressing cells (Figure 5D) and attenuated in PIN1-knockdown cells (Figure 5E). Additionally, MCF7 cells were pretreated with juglone, followed by treatment with or without IL-34. The results showed that juglone notably inhibited IL-34-induced phosphorylation of MEK1/2, ERK1/2, JNK1/2, and c-Jun (Figure 5F). Collectively, these results suggest that the IL-34-induced MAPK signaling pathway is regulated by PIN1 through CSF1R in MCF7 cells.

### 2.6. PIN1 Enhances IL-34-Induced AP-1 Activity and Breast Tumorigenesis

To further understand the role of IL-34 in breast tumorigenesis, we investigated the effects of IL-34 on AP-1 activity and anchorage-independent growth in MCF7 cells. Initially, we examined whether IL-34 could activate the c-Fos and c-Jun promoters in MCF7 cells. The transcriptional activities of c-Fos and c-Jun were significantly increased upon treatment with IL-34 in a dose-dependent manner (Figure 6A,B). 

Furthermore, treatment with IL-34 significantly induced AP-1 transactivation in MCF7 cells (Figure 6C). To further confirm that IL-34-induced AP-1 activation was regulated by PIN1, we assessed IL-34-induced AP-1 activity in siRNA-control and siRNA-PIN1-transfected MCF7 cells. As expected, silencing of PIN1 in MCF7 cells suppressed IL-34-induced transactivation activity of AP-1 (Figure 6D). Next, to study the effects and correlation of PIN1 activity and IL-34 on the tumorigenic potential of MCF7 cells resulting from increased AP-1 activity, we performed a soft agar colony formation assay and examined the effects of PIN1 depletion on IL-34-induced anchorage-independent growth of MCF7 cells. The number and size of colonies were found to be lower in PIN1-knockdown MCF7 cells, as compared to control cells (Figure 6E). Furthermore, the *in vivo* effects of juglone on IL-34-induced tumor development were studied in a BALB/c mice synergistic model with 4T1 cells. 4T1 cells were separately injected into the mammary glands of BALB/c mice in the presence or absence of IL-34 and juglone. Representative tumor images demonstrated that juglone significantly inhibited IL-34-induced mammary gland tumor development (Figure 6F). Altogether, our results indicate that PIN1 is important for IL-34-induced AP-1 activity and breast tumorigenesis.

## 3. Discussion

Breast cancer is the most common leading cause of cancer-related deaths in women worldwide [36]. The development and progression of cancer is closely associated with the process of inflammation [37]. Cancer cells in most solid tumors, including breast tumors, are surrounded by several types of inflammatory cells such as lymphocytes, TAMs, and natural killer cells, which together orchestrate a TME [1,2,3]. Inflammatory cells present in the TME release a wide range of soluble signaling molecules known as cytokines, which play an important role in the proliferation, survival, and metastasis of cancer cells [1]. ILs are one of the major cytokines present in the TME [6]. Previous studies have reported that ILs such as IL-17A, IL-22, and IL-33 promote neoplastic cell transformation and breast cancer tumorigenesis [32,38,39]. However, the role of IL-34 in breast cancer is still poorly understood. In the present study, we showed that IL-34 induces neoplastic cellular transformation and breast tumorigenesis through MEK/ERK and JNK/c-Jun signaling pathways. Furthermore, we illustrated that PIN1 enhances IL-34-induced phosphorylation of MEK1/2, ERK1/2, JNK1/2, and c-Jun. Overall, our study highlights the previously unknown role of IL-34/CSF1R signaling in breast cancer and demonstrates the regulatory role of PIN1 in IL-34-induced tumorigenesis.

IL-34 is a newly discovered cytokine that regulates the survival, differentiation, and function of macrophages, osteoclasts, and monocytes [34]. IL-34 expression is deregulated in inflammatory disorders and in response to infections [40]. Recently, IL-34 has been identified as a ligand for CSF1R that is involved in promoting disease progression in various conditions ranging from inflammation to cancer [21,41]. The engagement of IL-34 induces phosphorylation and activation of CSF1R, which subsequently triggers signaling cascades including NF-kB, ERK1/2, p38, JNK1/2, STAT3, AMPK/ULK1, and PI3K/Akt, which are associated with cell differentiation, survival, proliferation, migration, and motility [40]. Furthermore, interaction of IL-34 and CSF1R promotes the dramatic production of IL-6 by fibroblast-like synoviocytes through the JNK/P38/NF-κB signaling pathway [42]. Interestingly, CSF1R inhibition depletes TAMs, enriches tumor-infiltrating CD8+ T cells, and attenuates the growth of cervical cancer and mammary tumors [43]. Therefore, we hypothesized that IL-34/CSF1R signaling plays a role in epithelial cell transformation and breast tumorigenesis. In agreement with previous reports, we found that IL-34 strongly induced the phosphorylation of MEK1/2, ERK1/2, JNK1/2, and c-Jun, whereas knockdown of CSF1R considerably decreased these components of the MAPK pathway in JB6 Cl41 cells. Furthermore, treatment with PD98059 and SP600125, which are MEK1/2 and JNK1/2 inhibitors, strongly suppressed the IL-34-induced MEK/ERK and JNK/c-Jun signaling pathways in JB6 Cl41 cells. These data suggest that IL-34 activates MEK/ERK and JNK/c-Jun signaling through CSF1R and plays an important role in promoting tumorigenesis.

The MAPK signaling pathway regulates many fundamental cellular processes, such as cell transformation, differentiation, proliferation, and survival [44,45]. The MAPK pathway is activated by various stimuli such as growth factors, pro-inflammatory cytokines, and environmental stresses [46]. Activation of MAPKs causes the release of IL-6, IL-8, and TNF-α, which could lead to an inflammatory response [47]. In addition, the MAPK pathway plays an important role in the activation of activator protein 1 (AP-1) transcription factor, which is a dimeric complex comprised of members of the Fos, Jun, activating transcription factor and musculoaponeurotic fibrosarcoma protein families [48,49]. c-Jun is induced in human squamous cell carcinoma and has the potential to promote epidermal neoplasia [50]. c-Fos is a known oncogene expressed in sarcoma tumors, whose expression in immortalized mesenchymal progenitor cells results in cell transformation and chondrogenic tumor formation [51]. Previously, our group reported AP-1 as a major transcription factor in the transformation of JB6 Cl41 cells that is induced by various tumor promoters, such as epidermal growth factor and 12-O-tetradecanoylphorbol 13-acetate [52]. In addition, pro-inflammatory cytokines including IL-17, IL-33, and IL-22 have been shown to induce c-Fos and c-Jun activity, which results in neoplastic cell transformation through AP-1 signaling [32,38,39]. IL-34 expression has been observed in giant cell tumors of bone and human osteosarcoma cell lines and is associated with the progression of neoplasia [8,20]. However, the role of IL-34 in AP-1 activity and neoplastic cell transformation is largely unknown. This study provides evidence that IL-34 significantly induces c-Fos and c-Jun promoter activity, resulting in the induction of AP-1 transactivation activity in JB6 Cl41 cells. Moreover, treatment with IL-34 significantly increased the neoplastic transformation of JB6 Cl41 cells through activation of AP-1 activity.

A previous study has demonstrated the critical role of IL-34 in colon cancer cell proliferation and tumorigenesis [9]. In addition, IL-34 expression is enhanced in advanced stages of lung cancers and correlates with poor survival and disease progression in lung cancer patients [53]. There is also evidence that IL-34 plays a role in tumorigenesis given its ability to stimulate endothelial cell proliferation and recruitment of macrophages into the tumor tissue [8]. Although IL-34 plays an important role in cancer development, the molecular mechanism of IL-34 in breast cancer development has not yet been taken into consideration. In this study, we found that IL-34 triggers MAPK activation involving MEK1/2, ERK1/2, JNK1/2, and c-Jun through CSF1R in MCF7 cells. Consistent with these observations, we further showed that knockdown of CSF1R and/or treatment with PD98059 and SP600125 inhibited the IL-34-induced MEK/ERK and JNK/c-Jun pathways. Subsequently, IL-34 increased the tumorigenic potential of MCF7 cells through induction of c-Fos and c-Jun promoter activity. In addition, the *in vivo* mouse model study showed significant growth of tumors upon IL-34 treatment. These findings provide new insight into the molecular mechanisms underlying the tumor-promoting effects of IL-34 and suggest that the IL-34/CSF1R axis might be a potential therapeutic target to treat breast cancer.

The phosphorylation of proteins on serine or threonine residues that immediately precede proline (pSer/Thr-Pro) is specifically catalyzed by the prolyl isomerase PIN1 and is a key signaling mechanism in cell proliferation and transformation [54,55]. There is considerable evidence that PIN1 regulates the MAPK and STAT3 signaling pathways in breast tumor development through its interaction with MEK1 [29], c-Jun [30] and STAT3 [56]. In a previous report, the PIN1 inhibitor juglone has been shown to decrease the tumorigenicity of MCF7 breast cancer cells by inhibiting PIN1 activity [29]. These results support our hypothesis that PIN1 may regulate IL-34-induced MAPK signaling pathways leading to tumor development in breast cancer. The data in this report showed that knockdown of PIN1 strongly suppressed IL-34-induced phosphorylation of MEK1/2, ERK1/2, JNK1/2, and c-Jun in JB6 Cl41 and MCF7 cells. Furthermore, treatment with juglone considerably suppressed IL-34-induced AP-1 activity and cell transformation of JB6 Cl41 cells by inhibiting PIN1 activity. Silencing of PIN1 notably decreased IL-34-promoted tumorigenicity of MCF7 cells *in vitro*. We also used juglone to treat breast tumors in mice and found that juglone could decrease IL-34-induced *in vivo* breast tumor growth to a certain extent. Our findings indicate the previously unknown regulatory role of PIN1 in IL-34-induced cell transformation and tumorigenesis mediated by MEK/ERK and JNK/c-Jun pathways.

In conclusion, this study documents a formerly unknown role of IL-34 in human breast cancer tumorigenesis and epithelial cell transformation through its interaction with CSF1R. In addition, our study demonstrates the regulatory role of PIN1 in breast cancer tumorigenesis induced by IL-34. Moreover, the results of this study present PIN1 as an attractive therapeutic target in the TME, inhibition of which could potentially attenuate the aggressiveness of breast cancer. These findings may help establish a new therapeutic strategy against breast cancer.

## 4. Materials and Methods

### 4.1. Reagents and Antibodies

Recombinant mouse and human IL-34 (#5195-ML and #5265-IL) were obtained from R&D Systems (Minneapolis, MN, USA). The PIN1 inhibitor juglone (5-hydroxy-1,4-naphthoquinone) (#H47003) was obtained from Sigma-Aldrich (St. Louis, MO, USA). PD98059 (#513000) and SP600125 (#420119) were purchased from Calbiochem-Novabiochem (San Diego, CA, USA). Eagle’s minimal essential medium, L-glutamine, gentamicin, and fetal bovine serum (FBS) were purchased from Invitrogen (Carlsbad, CA, USA). Cell proliferation enzyme-linked immunosorbent assay and BrdU (colorimetric, #11647229001) were purchased from Roche Applied Science (Indianapolis, IN, USA). The dual-luciferase^®^ reporter assay kit (#E1980) was purchased from Promega (Madison, WI, USA). The cat. number and specificity of antibodies acquired from Cell Signaling Technology (Danvers, MA, USA) were as follows: MEK1/2 rabbit polyclonal antibody (#9122, detects a specific band for total MEK1/2 at 45 kDa), p44/42 MAPK (ERK1/2) (137F5) rabbit mAb (#4695, detects specific bands for total level of ERK1/2 at 42 and 44 kDa), SAPK/JNK (#9252, detects endogenous levels of total JNK1, JNK2 or JNK3 at 46 and 54 kDa), phospho-MEK1/2 (Ser217/221) (41G9) rabbit mAb (#9154, detects Ser217/221 phosphorylated MEK1/2 at 45 kDa), phospho-p44/42 MAPK (ERK1/2) (Thr202/Tyr204) (197G2) rabbit mAb (#4377, detects phosphorylated forms of p44 and p42 MAP kinases (ERK1 and ERK2) at 42 and 44 kD when dually phosphorylated at Thr202 and Tyr204 of ERK1 (Thr185 and Tyr187 of ERK2), and singly phosphorylated at Tyr204), phospho-SAPK/JNK (Thr183/Tyr185) (G9) mouse mAb (#9255, detects endogenous levels of p46 and p54 SAPK/JNK dually phosphorylated at Thr183 and Tyr185 at 46 and 54 kDa), phospho-c-Jun (Ser63) II rabbit polyclonal antibody (#9261, detect only the Ser63 phosphorylated form of c-Jun at 48 kDa). The antibodies acquired from Santa Cruz Biotechnology (Dallas, TX, USA) were as follows: c-Fms/CSF1R mouse mAb (C-20) (#sc-692, detects both unprocessed and processed forms of c-Fms/CSF1R at 130 and 165 kDa, respectively), PIN1 (G-8) mouse mAb (#sc-46660, detects a specific band at 20 kDa), c-Jun mouse mAb (G-4) (#sc-74543, detects a specific band at 48 kDa). The XPRESS mouse mAb (#R910-25, detects PIN1 fusion protein containing XPRESS epitope at 25 kDa) was acquired from Invitrogen. Anti-β-actin mouse mAb (#A1978, detects a specific band at 42 kDa) was obtained from Sigma-Aldrich.

### 4.2. Cell Culture and Transfection

JB6 Cl41 and MCF7 cells were cultured in Minimal Essential Medium (MEM) supplemented with 5% FBS and Dulbecco’s Modified Eagle Medium (DMEM) supplemented with 10% FBS, respectively. 4T1 mouse breast cancer cells were grown in Roswell Park Memorial Institute Medium (RPMI) supplemented with 10% FBS. All cell lines were cultured and maintained at 37°C in humidified air containing 5% CO_2_. DNA transfection into the cells was performed using jetPEI^®^, a cationic polymer transfection reagent (Polyplus-transfection, New York, NY, USA). Human PIN1 (accession number: NM_006221), mouse PIN1 (accession number: NM_023371), human CSF1R (accession number: NM_005211), and mouse CSF1R (accession number: NM_001135100) were silenced by transfecting with ON TARGETplus small interfering RNA (siRNA) SMARTpool-specific or non-specific control pool double-stranded RNA oligonucleotides (Dharmacon, Chicago, IL, USA) using Lipofectamine™ 2000 (Invitrogen).

### 4.3. Cell Proliferation Assay via 5-Bromo-2’-Deoxyuridine (BrdU) Incorporation

Cells were seeded (5 × 10^4^ cells/well) into 96-well plates in 100 µL of MEM, DMEM, and RPMI supplemented with 5% or 10% FBS, as appropriate. After 24 h, the cells were treated with or without IL-24 for 48 h, labeled with 10 µL/well BrdU labeling solution, and then incubated for 4 h at 37 °C in a 5% CO_2_ atmosphere. Cell proliferation was estimated by measuring the absorbance at 370 nm.

### 4.4. Immunoblot Analysis

The cells were disrupted in radioimmunoprecipitation assay lysis buffer. The proteins were resolved using sodium dodecyl sulfate-polyacrylamide gel electrophoresis and transferred onto polyvinylidene difluoride membranes. The membranes were blocked and hybridized with the appropriate primary antibody overnight at 4 °C. After hybridization with HRP (horseradish peroxidase)-conjugated secondary antibody from rabbits or mice, the protein bands were visualized using a chemiluminescence detection kit (HRP Chemiluminescent Substrates, Amersham Biosciences, Piscataway, NJ, USA). LAS4000 system (GE Healthcare Biosciences, Pittsburgh, PA, USA) was used for chemiluminescence detection.

### 4.5. Anchorage-Independent Cellular Transformation Assay (Soft Agar Assay)

The effect of IL-34 on cell transformation was investigated in JB6 Cl41 and MCF7 cells. Briefly, 8 × 10^3^ cells were exposed to different doses of IL-34 in 1 mL of 0.3% basal medium eagle containing 10% FBS, 2 mM l-glutamine, and 25 µg/mL gentamicin. The cultures were maintained at 37 °C in a 5% CO_2_ incubator for 14–18 d, following which the cell colonies were scored using an Axiovert 200M fluorescence microscope and AxioVision software (both from Carl Zeiss, Thornwood, NY, USA).

### 4.6. Reporter Gene Assay

To detect firefly luciferase activity, a reporter gene assay was performed using lysates from AP-1-, c-Jun-, and c-Fos-luc-transfected JB6 Cl41 and MCF7 cells. The reporter gene vector pRL-TK-luciferase plasmid (Promega) was co-transfected into each cell line and the Renilla luciferase activity generated by this vector was used to normalize the results for transfection efficiency. Cell lysates were mixed with Luciferase Assay Reagent II, following which firefly luciferase light emission was measured using GloMax^®^-Multi Detection System (Promega). Subsequently, Renilla luciferase substrate was added to this setup to normalize the firefly luciferase data. The c-Fos-luc (pFos-WT GL3) and c-Jun-luc (JC6GL3) promoter constructs were kindly provided by Dr. Ron Prywes (Columbia University, New York, NY, USA). The AP-1 luciferase reporter plasmid (−73/+63 collagenase-luciferase) was kindly provided by Dr. Dong Zigang (Hormel Institute, University of Minnesota, Austin, MN, USA).

### 4.7. Tumorigenicity Assay in BALB/c Mice

Six-week-old female BALB/c mice (18–20 g) were obtained from Samtako Co. (Osan, Republic of Korea), acclimatized for 1 week, and maintained in a clean room at the College of Pharmacy, Chosun University (Gwangju, Korea). The animals were caged under filtered pathogen-free air at a temperature between 20 °C and 23 °C with a 12 h/12 h light/dark cycle and a relative humidity of 50%. The animals were fed commercial rat chow (OrientBio, Co., Seongnam, Korea) and had access to water ad libitum. The protocols of the animal studies were approved by the Animal Care Committee of Chosun University. Mice were randomly divided into three groups of 10 animals each. 4T1 mouse breast cancer cells derived from BALB/c mice were then injected into the mammary glands of the mice in the presence or absence of IL-34 and allowed to grow until formation of tumors (14 d). The mice were observed daily for tumor growth. Tumor volume was calculated using the formula: V = (ab^2^)/2, where ‘a’ was the longest diameter and ‘b’ was the shortest diameter of the tumor.

### 4.8. Statistical Analysis

Data from the reporter gene assay and soft agar assay were analyzed statistically using one-way ANOVA and *p* values < 0.05 were considered significant. Statistical calculations were carried out using Prism 8.4.2 (GraphPad Software, LLC). Results have been expressed as mean ± standard error of triplicate measurements from three independent experiments.

## Figures and Tables

**Figure 1 ijms-22-02711-f001:**
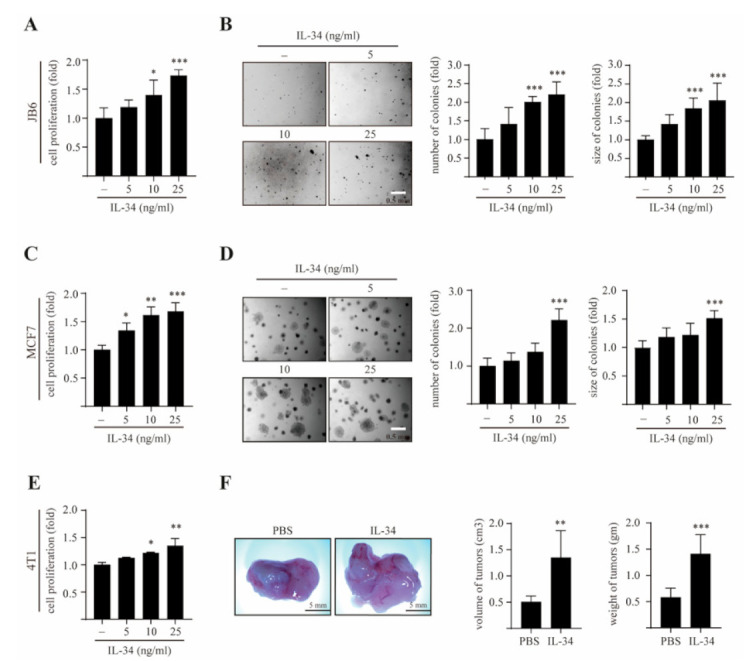
Effects of IL-34 on anchorage-independent growth and mammary gland tumorigenesis in vitro and in vivo. (**A**) JB6 Cl41 cells were treated with different concentration of IL-34 for 48 h, following which the cell proliferation was estimated using BrdU incorporation assay. (**B**) JB6 Cl41 cells were exposed to different concentrations of IL-34 as indicated, subjected to soft agar assay, and incubated at 37 °C in a 5% CO_2_ atmosphere for 14 days. Representative colonies from three separate experiments were photographed (left), followed by calculation of the average colony numbers and sizes (right). (**C**) MCF7 cells were treated with various doses of IL-34 for 48 h and cell proliferation was approximated using BrdU incorporation assay. (**D**) MCF7 cells were treated with the indicated concentration of IL-34 in a soft agar matrix and incubated at 37 °C in a 5% CO_2_ atmosphere. After 14 days, colonies from three separate experiments were photographed (left), followed by calculation of the average colony number and sizes (diameter > 100 μm, right). (**E**) 4T1 cells were seeded and treated with different concentrations of IL-34 for 48 h. Cell proliferation was then estimated using BrdU incorporation assay. (**F**) 4T1 cells were injected into the mammary gland of BALB/c mice in the presence or absence of 100 ng/mL IL-34 and allowed to grow until tumors were formed. Shown are representative pictures of tumor (left), measured volumes and weights of tumors (right). Error bars indicate mean ± S.D. of triplicate measurements from three independent experiments. Statistical analyses were conducted using one-way ANOVA (**p* < 0.05, ** *p* < 0.01, *** *p* < 0.001, compared to the control groups).

**Figure 2 ijms-22-02711-f002:**
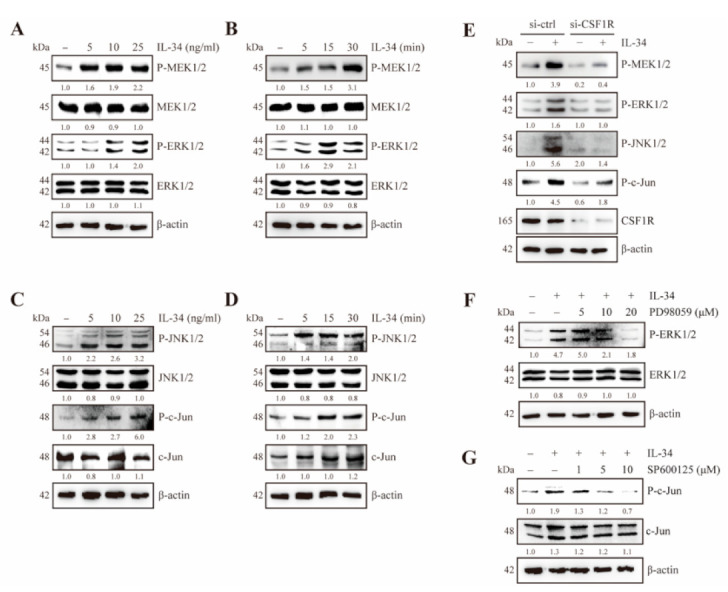
Effects of IL-34 on MEK/ERK and JNK/c-Jun signaling pathways in JB6 Cl41 cells. (**A**–**D**) Cells were serum starved for 24 h, treated with indicated doses of IL-34 for 30 min (A,C) or 10 ng/mL IL-34 for the indicated times (**B**,**D**), harvested, and lysed. The lysates were resolved using SDS-PAGE and immunoblotting analysis was performed using specific antibodies against corresponding proteins. (**E**) Cells were transfected with mouse siRNA-control and siRNA-CSF1R. At 24 h after transfection, the cells were serum starved for 24 h, treated with 10 ng/mL IL-34 for 15 min or left untreated, harvested, and lysed. Proteins in whole cell lysates were separated using SDS-PAGE and immunoblotted. (**F**,**G**) Cells were serum starved for 24 h, pre-treated with different concentrations of PD98059 (**F**) or SP600125 (**G**) for 12 h, exposed to 10 ng/mL IL-34 for 15 min, harvested, and lysed. Proteins in whole cell lysates were separated using SDS-PAGE and immunoblotted. (**A**–**F**) Blots are representative of an experiment repeated at least three times with the similar result. The numbers below the band represent fold changes in protein levels after normalization to β-actin using densitometric quantification by ImageJ.

**Figure 3 ijms-22-02711-f003:**
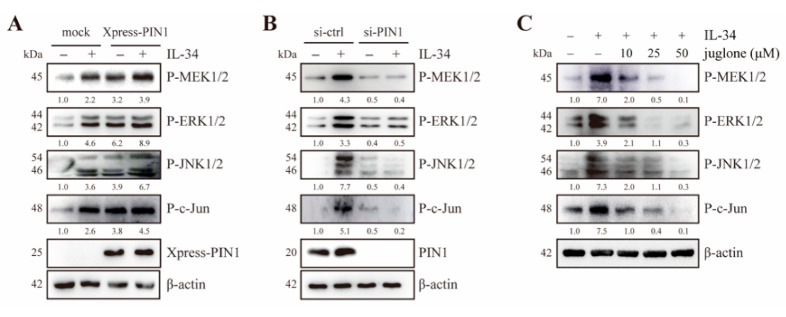
Role of PIN1 in IL-34-induced MEK/ERK and JNK/c-Jun signaling pathways in JB6 Cl41 cells. (**A**) Cells were transfected with mock and Xpress-PIN1. At 24 h after transfection, the cells were serum starved for 24 h, treated with 10 ng/mL IL-34 for 15 min or left untreated, harvested, and lysed. The lysates were resolved using SDS-PAGE and immunoblotting analysis was performed using specific antibodies against corresponding proteins. (**B**) Cells were transfected with mouse siRNA-control and siRNA-PIN1. Cells were serum starved for 24 h, treated with 10 ng/mL IL-34 for 15 min or left untreated, harvested, and lysed. Proteins in whole cell lysates were separated using SDS-PAGE and immunoblotted. (**C**) Cells were serum starved for 24 h, pre-treated with the indicated concentrations of juglone for 12 h, and then exposed to 10 ng/mL IL-34 for 15 min, harvested, and lysed. Proteins in whole cell lysates were separated using SDS-PAGE and immunoblotted. Blots are representative of an experiment repeated at least three times with the similar result. The numbers below the band represent fold changes in protein levels after normalization to β-actin using densitometric quantification by ImageJ.

**Figure 4 ijms-22-02711-f004:**
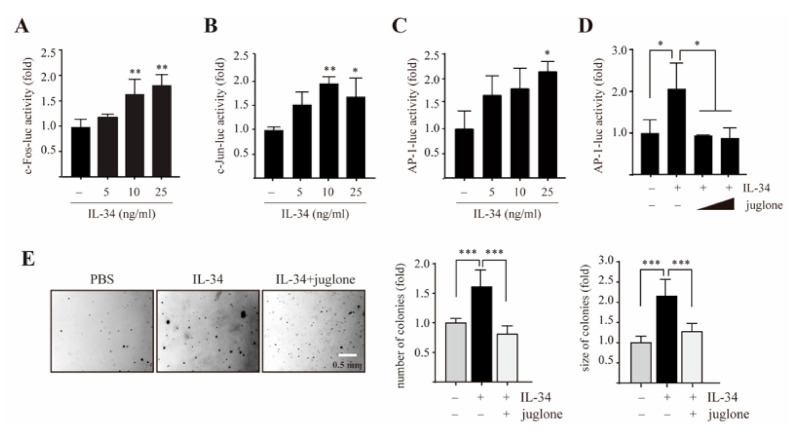
Effects of juglone on IL-34-induced AP-1 activity and cell transformation in JB6 Cl41 cells. Cells were seeded and co-transfected with the luciferase reporters, c-Fos-luc (**A**), c-Jun-luc (**B**), AP-1-luc (**C**) along with pRL-TK vector. At 24 h after transfection, the cells were serum starved for 24 h, and then treated with the indicated doses of IL-34 for 24 h before a luciferase assay was performed. (**D**) Cells were seeded and co-transfected with the luciferase reporter AP-1-luc and pRL-TK vector. After 24 h, the cells were serum starved for 24 h, pretreated with the indicated doses of juglone and then exposed to 10 ng/mL IL-34 for 24 h before a luciferase assay was performed. (**E**) Cells were treated with 10 ng/mL IL-34 in the presence or absence of juglone in a soft agar matrix and incubated at 37 °C in a 5% CO_2_ atmosphere for 14 days. Representative colonies from three separate experiments were photographed (left), followed by calculation of the average colony numbers and sizes (diameter > 200 μm, right). Data in (**A**–**E**) represents the mean ± S.D. of triplicate measurements from three independent experiments. Statistical analyses were conducted using one-way ANOVA (**p* < 0.05, ** *p* < 0.01, *** *p* < 0.001, compared to control group or only IL-34-treated group).

**Figure 5 ijms-22-02711-f005:**
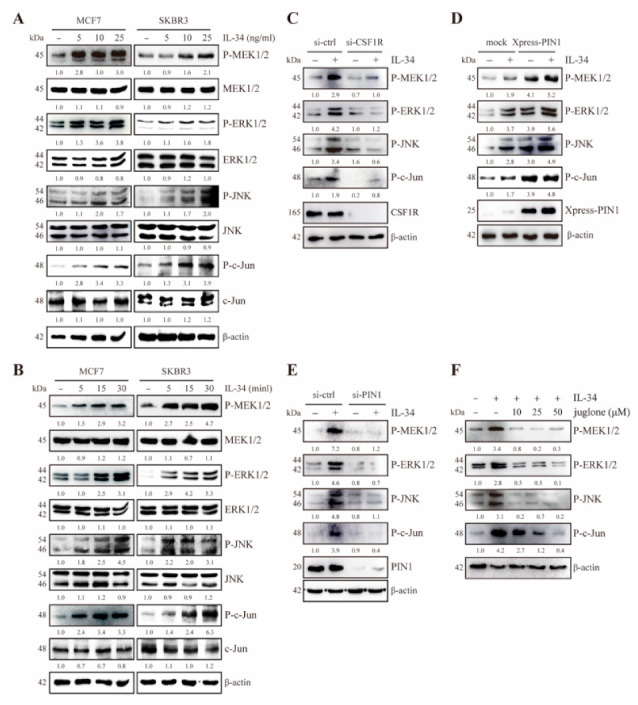
Involvement of PIN1 in IL-34-induced MAPK pathway in breast cancer cells. (**A**,**B**) SKBR3 and MCF7 cells were serum starved for 24 h, treated with the indicated doses of IL-34 for 30 min (**A**) or 10 ng/mL IL-34 for indicated times (**B**), harvested, and lysed. The lysates were resolved using SDS-PAGE and immunoblotting analysis was performed using specific antibodies against corresponding proteins. (**C**–**E**) MCF7 cells were transfected with siRNA-control and siRNA-CSF1R (**C**) mock and Xpress-PIN1 (**D**) or siRNA-control and siRNA-PIN1 (**E**), serum starved for 24 h and then treated with 10 ng/mL IL-34 for 15 min or left untreated, harvested, and lysed. Proteins in whole cell lysates were separated using SDS-PAGE and immunoblotted. (**F**) MCF7 cells were serum starved for 24 h, pre-treated with the indicated concentrations of juglone for 12 h, and then exposed to 10 ng/mL IL-34 for 15 min, harvested, and lysed. Proteins in whole cell lysates were separated using SDS-PAGE and immunoblotted. (**A**–**F**) Blots are representative of an experiment repeated at least three times with the similar result. The numbers below the band represent fold changes in protein levels after normalization to β-actin using densitometric quantification by ImageJ.

**Figure 6 ijms-22-02711-f006:**
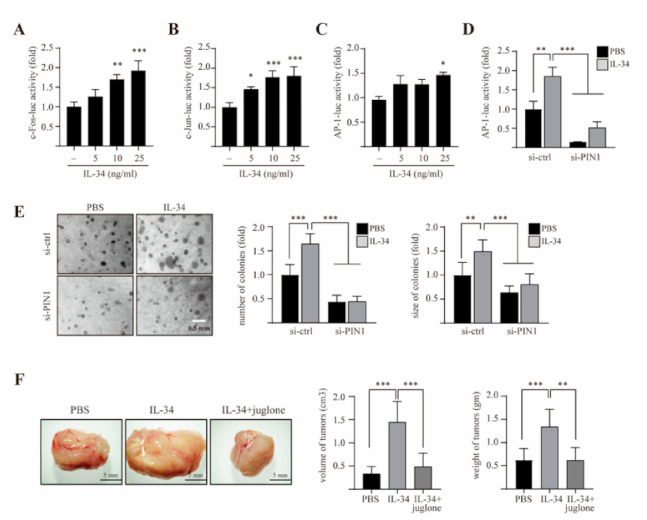
Role of PIN1 in AP-1 activity and breast tumorigenesis induced by IL-34. MCF7 cells were seeded and co-transfected with the luciferase reporters, c-Fos-luc (**A**), c-Jun-luc (**B**), and AP-1-luc (**C**) along with the pRL-TK vector. At 24 h after transfection, the cells were serum starved for 24 h and then treated with the indicated doses of IL-34 for 24 h before luciferase assay was performed. (**D**) MCF7 cells were seeded and co-transfected with the luciferase reporter AP-1-luc and siRNA-control or AP-1-luc and siRNA-PIN1. At 24 h after transfection, the cells were serum starved for 24 h, treated with 10 ng/mL IL-34 for 24 h or left untreated, before a luciferase assay was performed. (**E**) MCF7 cells were transfected with siRNA-control and siRNA-PIN1. After 24 h, the cells were treated with 10 ng/mL IL-34 in a soft agar matrix or left untreated and incubated at 37 °C in a 5% CO_2_ atmosphere. After 14 days, colonies from three separate experiments were photographed (left), followed by calculation of the average colony numbers and sizes (diameter > 200 μm, right). (**F**) 4T1 cells were injected into the mammary gland of BALB/c mice in the presence or absence of 100 ng/mL IL-34 and 100 μM juglone, and allowed to grow until tumors were formed. Shown are representative pictures of tumor (left), measured volumes and weights of tumors (right). Error bars indicate the mean ± S.D. of triplicate measurements from two independent experiments. Statistical analyses were conducted using one-way ANOVA (**p* < 0.05, ** *p* < 0.01, *** *p* < 0.001, compared to control group or only IL-34-treated group, respectively).

## Data Availability

The data presented in this study are available on request from the corresponding author.

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
