# Peer review of "Interleukin-34-CSF1R Signaling Axis Promotes Epithelial Cell Transformation and Breast Tumorigenesis"

_ijms, 2021, doi:10.3390/ijms22052711_

Round 1

Reviewer 1 Report

In this paper, the authors describe clearly and concisely the regulation of MEK/ERK and JNK/c-Jun pathways by Interleukin 34/CSF1R interplay in mouse skin and human mammary cell lines. They also highlight the regulatory role of   peptidyl-prolyl 23 cis-trans isomerase NIMA-interacting 1 (PIN1) in the promotion of epithelial cell transformation and breast tumoral phenotipe sustained by IL34/CSF1R interaction. Consequently, they propose PIN1-targeted therapy as a possible tool to counteract breast cancer progression.

Few points must be considered:

  1. Line 76: In my opinion, it could be useful to introduce PIN1 with a brief explanation of its functional role in the general context, i.e.: Peptidyl-prolyl 23 cis-trans isomerase NIMA-interacting , or PIN1, is an enzyme which regulates....
  2. 1 -Panel D: In order to understand the size of different colonies, it would be useful the presence of a size bar.
  3. 1 -Panel D: In MCF7, Il-34 treatment increased cell proliferation; the number of colonies did not change, but at higher doses, their dimensions did. How this data could be explained and fit with the statement: IL34 induces.....and breast mammary tumorigenesis in vitro? (Line 133).
  4. Line 126: It could be useful to add “a mouse mammary cell line derived from BALB/c strain” after 4T1.
  5. Line 186: It could be useful to add “(5-Hydroxy-1,4-naphthoquinone)” after PIN1.
  6. Line 196: “at the higher dose”, instead of "in a dose-dependent manner", as it is evident in Fig. 4C.
  7. Line 220: “influences” instead of "is important" (Line 220)
  8. 3A: IL-34 increases the phosphorylated forms of MEK1/2, ERK1/2, JNK1/2, and c-Jun in MCF7 cells, but in SKBR3 cells the densitometric value of P-c-Jun seems not corresponding to the image (Fig. 3A).

Reviewer 2 Report

The manuscript is well and clearly written, experiments are described well, and there is a solid logic in how data is described and discussed. The few questions and comments I have are the following:

1) Lines 36-37: Authors are citing research on melanoma for IL-1, IL-4, IL-6 effects on tumor formation but there is a lot of data published on the effect of these cytokines in breast cancer; citing those might be more relevant for this research.

2) Figure 1 and others: If authors are willing to indicate “0” instead of “-“ for the controls on the diagrams it might be more demonstrative.

3) Animal experiments results need more explanation. When describing this experiment, in line 479 authors indicate “presence or absence of IL-34 and juglone”. Does it mean that BC cells were treated with IL-34 and/or juglone once (before getting injected)? Why was this course of treatment chosen? It’s more common when treatments are administered to mice every 1/3/5 days. The results shown are significant; however, do authors conclude that IL-34 and/or juglone was preserved in tumor microenvironment for 14 days? (in this case, what is half-life for IL-34 and juglone in vivo?) Or the original effect of IL-34 and/or juglone is enough to cause long-term effects on the cells proliferation in vivo? There is no data on cell signaling changes in tumors derived from mice, so it’s hard to apply conclusions from cell culture experiments towards in vivo situation. The colony formation assay shows tumor cells growth decrease by IL-34 but it’s not the same conditions, because IL-34 was present in the media during whole experiment.

Reviewer 3 Report

The work presented by Poudel et al. is well written and discussed. The experimental design has been extensively described and the methodological approach used is appropriate. However, some considerations to improve the manuscript have to be pointed:

  • The introduction looks too long and confusing. It should be reduced focusing of main treated aspects, while the accurate descriptions of pathways should be moved in the discussion.
  • I suggest to confirm the IL-34 induced-pathways on tissues derived by mice models treated with IL-34 and IL-34+juglone.

Round 2

Reviewer 2 Report

The authors provided explanation for all reviewer's questions.